# RSV Reprograms the CDK9•BRD4 Chromatin Remodeling Complex to Couple Innate Inflammation to Airway Remodeling

**DOI:** 10.3390/v12040472

**Published:** 2020-04-22

**Authors:** Allan R. Brasier

**Affiliations:** Institute for Clinical and Translational Research; University of Wisconsin-Madison School of Medicine and Public Health; Madison, WI 53705, USA; abrasier@wisc.edu

**Keywords:** Airway remodeling, cyclin dependent kinase 9 (CDK9), bromodomain containing protein 4 (BRD4), mesenchymal transition, myofibroblast, epigenetics

## Abstract

Respiratory syncytial virus infection is responsible for seasonal upper and lower respiratory tract infections worldwide, causing substantial morbidity. Self-inoculation of the virus into the nasopharynx results in epithelial replication and distal spread into the lower respiratory tract. Here, respiratory syncytial virus (RSV) activates sentinel cells important in the host inflammatory response, resulting in epithelial-derived cytokine and interferon (IFN) expression resulting in neutrophilia, whose intensity is associated with disease severity. I will synthesize key findings describing how RSV replication activates intracellular NFκB and IRF signaling cascades controlling the innate immune response (IIR). Recent studies have implicated a central role for *Scg1a1^+^* expressing progenitor cells in IIR, a cell type uniquely primed to induce neutrophilic-, T helper 2 (Th2)-polarizing-, and fibrogenic cytokines that play distinct roles in disease pathogenesis. Molecular studies have linked the positive transcriptional elongation factor-b (P-TEFb), a pleiotrophic chromatin remodeling complex in immediate-early IIR gene expression. Through intrinsic kinase activity of cyclin dependent kinase (CDK) 9 and atypical histone acetyl transferase activity of bromodomain containing protein 4 (BRD4), P-TEFb mediates transcriptional elongation of IIR genes. Unbiased proteomic studies show that the CDK9•BRD4 complex is dynamically reconfigured by the innate response and targets TGFβ-dependent fibrogenic gene networks. Chronic activation of CDK9•BRD4 mediates chromatin remodeling fibrogenic gene networks that cause epithelial mesenchymal transition (EMT). Mesenchymal transitioned epithelial cells elaborate TGFβ and IL6 that function in a paracrine manner to expand the population of subepithelial myofibroblasts. These findings may account for the long-term reduction in pulmonary function in children with severe lower respiratory tract infection (LRTI). Modifying chromatin remodeling properties of the CDK9•BRD4 coactivators may provide a mechanism for reducing post-infectious airway remodeling that are a consequence of severe RSV LRTIs.

## 1. Impact and Sequelae of RSV Infections

Respiratory syncytial virus (RSV) is responsible for seasonal outbreaks of respiratory tract infections worldwide. This enveloped, single-stranded, negative-sense RNA virus of the *Orthopneumovirus* genus of the Pneuomoviridae family is the most common cause of pediatric hospitalization in children less than five years of age [1]. RSV is spread by large droplet spread and self-inoculation of the nasopharynx. RSV attaches, fuses, and replicates in its primary epithelial cell target. Afterwards, infectious RSV virions spread to the small bronchiolar airway epithelium by cell-to-cell spread or by inhalation of secretions [2]. In immunologically naïve, or immunosuppressed individuals, RSV spread into the lower airways produces lower respiratory tract infection (LRTI), whose clinical features include bronchiolitis and pneumonia. Pathologically, LRTI is associated with epithelial giant cell formation, necrosis, sloughing, producing mucous plugging, ventilation-perfusion mismatching, and acute hypoxic respiratory failure [2,3,4]. The findings that the initial clinical manifestations of hypoxia are correlated with high viral titers and epithelial-derived cytokines [5,6] indicates that RSV activation of the IIR plays an important component of early disease manifestations.

Observational studies of naturally occurring RSV infections in humans suggest that acute infection produces an initial neutrophilic airway inflammation [7], followed by a CD8+ T-cell response important in viral clearance (reviewed in [8]). Because protective IgA antibodies wane six months after infection, re-infection occurs throughout life. A focus of the investigations of naturally occurring disease has been on understanding why the adaptive response to RSV is transient. These defects may be due to defective formation of memory CD8+ T-cells [9], suppression of activated CD8+ T-cells via PD-L1 [10], or induction of CD8+ T-cell apoptosis [8].

Complementing these observational studies, adult challenge studies, where the timing and inoculum can be precisely controlled, have provided unique insights into disease pathogenesis [11]. After inoculation, clinical signs and symptoms of upper respiratory tract infection became manifest, and peak virus shedding occurred seven days after inoculation, and viral clearance from the respiratory secretions was largely complete after 15 days. In a subset of infected volunteers, a strikingly extensive macroscopic inflammation of the lower respiratory tract was seen by bronchoscopy. Here, viral antigen, giant cell formation, and mucosal epithelial sloughing persisted up to 28 days later, although lower respiratory tract symptoms were largely absent. These findings suggest that even in adults with prior RSV exposure, upper respiratory tract infections (URIs) are associated with asymptomatic viral replication and persistent mucosal inflammation in the lower tract.

## 2. The Airway Epithelial Cell as a Sensor of RSV Replication

Airway epithelial cells are poised for detecting and dynamically responding to viral attack through an arsenal of pattern recognition receptors (PRRs) monitoring the airway lumen, cellular cytoplasm, and subcellular organelles for the presence of pathogen associated molecular patterns (PAMPs; [12,13]). Luminal viral PAMPs, double-stranded RNA (dsRNA) and 5-phosphorylated RNA, are primarily bound by membrane-associated Toll-like receptor 3 (TLR3) present on airway epithelial cells. In contrast, intracellular viral PAMPs are detected by dsRNA helicases and kinases of the retinoic acid inducible gene -I (RIG-I)/melanoma differentiation-associated protein (MDA5) family. Upon binding their cognate PAMPs, PRR-mediated signaling produces acute oxidative injury [14], disruption of cilia function, epithelial apoptosis, disruption of epithelial barrier function [15], and activation of IIR signaling, resulting in rapid neutrophilic inflammation [16]. These pathways have been extensively reviewed [17] and modeled mathematically [18].

The initial events in RSV sensing are well-understood. After viral binding and fusion [19], RSV genomic RNA and its nucleoprotein complex including RNA-dependent RNA polymerase are released into the cytoplasm, where the polymerase produces capped and polyadenylated mRNAs encoding viral proteins in cytoplasmic inclusion bodies [20], becoming sites of viral RNA synthesis and production of innate immune responses. In epithelial cells, UV cross-linking immunoprecipitation PCR experiments have demonstrated that RSV RNA is primarily sensed by RIG-I [21]. RIG I is a helical ATP-dependent DExD/H box RNA helicase that binds to 5’phosphorylated RNA and short dsRNA produced during RSV transcription [22]. RIG-I appears to be the primary initial sensor of RSV molecular patterns, although MDA5 may be important in immune cells [23]. The caspase activation and recruitment domain (CARD) of RIG-I binds to CARDs of the mitochondrial antiviral signaling protein (MAVS) to activate formation of the inflammasome on the mitochondrial surface. The inflammasome is a signaling receptor scaffold that recruits and activates kinases and ubiquitin ligases that are rate limiting in the activation and nuclear translocation of the interferon regulatory factor (IRF)3 and nuclear factor-κB (NFκB) transcription factors.

NFκB is a family of cytoplasmic transcription factors activated by RIG-I in a two-step process: (1) liberation from cytoplasmic inhibition, and (2) Transactivation [24,25]. It is well-established that calpain and proteasome pathways mediate cytoplasmic proteolysis of IκB inhibitors to release NFκB from cytoplasmic stores [26], resulting in exposure of nuclear localization sequences promoting efficient translocation of NFκB into the nucleus. However, we made the seminal observation that nuclear translocation alone is insufficient to activate innate NFκB-dependent networks [27,28]. The mechanism underlying viral induced NFκB activation is via the production of reactive oxygen species (ROS)-induced nuclear damage. ROS-induced DNA damage (guanine oxidation) is transduced through nuclear export of the ataxia telangiectasia mutated (ATM) kinase culminating in mitogen and stress kinase (MSK)-1 activation and RelA phosphorylation ([28]; reviewed in [14]). This additional requirement of nuclear oxidative stress response signaling to license full activation of the IIR ensures that this response is only activated by active replicating pathogens, and not inert (nonreplicating) inhaled PAMPs.

## 3. Activation and Cross-Talk of the NFκB and IRF3 Pathways in Response to RSV

In epithelium, IRF-1, -3, and -7 are the key regulators of type I and type III IFN gene expression [29], responsible for expression of some 350 interferon simulating genes (ISGs), producing anti-viral protection [30]. IFN type I/III stimulation produces a dramatic upregulation of RIG-I to further amplify the anti-viral response. IFN stimulation upregulates TLR-3 and -4 expression and cause them to redistribute from sequestered endosomal compartments to the cell surface [21]. In a similar manner, IFN induces LRR-, and CARD-containing 5 (NLRC5), priming RSV infected cells to respond to additional PAMPs [31]. Through these effects, IFN auto-amplification primes neighboring epithelial cells to elicit a protective anti-viral state and commitment to apoptosis [32].

An important discovery has been the finding that NFκB and IRF3 signaling modules have extensive cross-talk in a cell-type restricted pattern. Although IRF3 activation is largely NFκB-independent, sustained IRF action on IFN and ISG expression requires ongoing NFκB signaling in epithelial cells. Some key elements of NFκB-IRF pathway cross-talk are that: (1) the virus infection-activated IFNβ enhanceosome is activated by NFκB (note that IFNβ is a major type I IFN, responsible for the majority of mucosal anti-viral effect); (2) the induction of inducible IRF-1 and -7 subunits controlling type I IFN expression are dependent on direct NFκB transactivation [33]; and, (3) NFκB is required for the IRF-IFN-RIG-I amplification loop, described above [34]. Consequently, sustained epithelial IFN response is substantially blunted in the absence of NFκB [33,34].

## 4. The Role of the Airway Epithelium in Innate Responses to RSV

The airway epithelium provides a major role in pulmonary host defense through multiple mechanisms [13]. Although studies have shown an important modulatory role of macrophages in RSV induced disease [35], the data that the epithelium plays a role in initial RSV anti-viral IIR and disease pathogenesis are compelling. In natural infections, RSV replicates to high levels in airway epithelial cells of the nasal, tracheal, and lower airways, including type I and II alveolar epithelial cells [2,3]. It is well established that RSV-infected epithelial cells rapidly secrete cytokines [36], interferons [21], exosomes [37], and damage-associated patterns [38]. Many of these epithelial specific cytokines, such as type III interferons (IFNs), IL-33, MCP, and MIP1α, can be detected in the airway fluids of naturally infected children with LRTI and experimentally challenged adults [6,9,39,40]. Systems level studies of epithelial responses have shown that RSV infection produces global genomic reprogramming [36], an event coupled to changes in secreted protein (secretome) profiles.

The nature, timing, and magnitude of inducible epithelial cytokine responses play important roles in shaping the evolution of adaptive immunity. As examples, epithelial CXCL1/IL8 is a major chemoattractant for the initial neutrophil response, cells that mediate phagocytosis and express protective neutrophil extracellular traps (NETs). Epithelial type I and III IFNs play an important role in anti-viral response of neighboring epithelial cells. Damage-associated HMGB1 is important in monocyte recruitment and activation [41]. Epithelial B-cell activating factor (BAFF) is responsible for T-cell independent B-cell activation mediating the pulmonary antibody response [42]. Expression of Th2-activating cytokines MIP, CCL5, TSLP, and others influence the propensity for developing atopy. These are a few examples of diverse responses directed by RSV infected epithelium.

## 5. NFκB Signaling in Bronchiolar Cells Generate Unique Innate Signals Important in LRTI Pathogenesis

With respect to the IIR, the airway epithelium can be divided into three anatomically distinct regions: trachea, bronchioles, and alveoli. Comparison of the IIR mechanisms employed by epithelial cells from these regions shows marked phenotypic diversity [37]. Ciliated and secretory tracheal epithelial cells provide innate defenses by mucociliary escalator activity and mucin secretion that bind pathogens and facilitates their clearance. Nonciliated bronchiolar epithelial cells inducibly synthesize and secrete cytokines as free or nanoparticles (exosomes) that stimulate inflammation and promotes an anti-viral and pro-apoptotic state. Alveolar epithelial cells also produce cytokines and produce surfactant that promotes macrophage opsonization. 

Gene-profiling experiments have suggested that lower airway epithelial cells produce greater amounts of T helper type 2 (Th2)-activating CCL-type chemokines than do epithelial cells of the conducting airways [20]. These findings were extended by recent systems-level proteomics studies that have shown cell-type differences of inducible secreted proteins (secretome) in response to RSV infection. Using a highly sensitive unbiased secretome profiling technique, we observed that airway epithelial cells dynamically respond to RSV infection by synthesizing and secreting ~1000 proteins, both as free and membrane-bound nanoparticles (exosomes) [37]. These common proteins were derived from lysosomal, cytoplasmic, and nuclear compartments. Interestingly, many of these secreted proteins do not contain classic signal peptides, suggesting novel and incompletely understood mechanisms for innate signaling. For example, we have demonstrated that the nuclear damage-associated molecular patterns (DAMPs), HMGB1, and histone H3 undergo nuclear export and extracellular secretion in response to RSV; these DAMPs controlling mononuclear inflammation [38,41]. Informatics analysis indicated that RSV infected *Scgb1a1*-expressing bronchiolar cells selectively secrete 103 unique proteins and exosomal contents (Cluster 3 in Figure 1C). Of these unique proteins, Th2 activating cytokines (MIP1, TSLP) [40], mucin expressing (CCL20), and fibrogenic cytokines (IL6) were identified. Importantly, all of these factors are dependent on NFκB, further implicating NFκB in immunopathogenesis and remodeling in RSV LRTI.

## 6. Unique RSV-Induced Bronchiolar Cell Factors Mediate LRTI Disease

The identification of differential protein secretion patterns by lower airway cells is highly relevant to the pathogenesis of RSV disease. Human cohort studies have observed that RSV LRTIs are associated with Th2 polarization and enhanced aeroallergen sensitivity [43]. TSLP is a mediator of RSV-induced airway inflammation whose secretion by RSV-infected airway cells occurs in an NFκB–dependent mechanism [44]. Although TSLP has a broad variety of target cell responses, its ability to activate plasmacytoid dendritic cell populations and promote Th2 lymphocyte–predominant inflammation is especially relevant to the pathophysiology of severe LRTI infection. Mucous production and small airways obstruction is a hallmark of fatal RSV infection [4]. Our secretome analysis demonstrated that CCL20 is the major mucin-producing factor secreted from RSV-infected hSAECs. Bronchiolar secretion of CCL20 also promotes plasmacytoid Dendritic Cell (pDC) and Th17 lymphocyte recruitment to sites of inflammation [45]. Th17-mediated inflammation has been implicated in the pathogenesis of RSV LRTIs [46], such that children with RSV LRTI have increased IL-17 levels in their airway fluids. Finally, IL6 is an NFκB-responsive gene induced by RSV that plays an important paracrine role on myofibroblast transdifferentiation, discussed later.

## 7. Bronchiolar NFκB Signaling Mediates RSV Disease In Vivo

To study the role of nonciliated small airway cells on RSV-induced inflammation in vivo, we developed an inducible NFκB/RelA knockout mouse, where RelA could be deleted in small airway epithelial cells [47,48]. Mice depleted of RelA in the *Scgb1a1* progenitor-derived population have significantly reduced chemokine response, leukocytic inflammation, and airway obstruction in response to experimental RSV infection [37]. Interestingly, TLR3-driven inflammation is also mediated by the same epithelial cell population [48]. In a manner similar to the findings with RSV infection, mice depleted of RelA in the *Scgb1a1^+^* progenitor cells respond to luminal TLR3 agonist with reduced epithelial-dependent chemokine expression, neutrophilia, airway hyperreactivity, and remodeling. Collectively these data are consistent with the notion that NFκB signaling in this bronchiolar cell population is a major mediator of viral induced lung inflammation downstream of RIG-I or TLR3 responsible for secreting pathogenic factors controlling disease.

## 8. NFκB Mediates the IIR by Transcriptional Elongation

The central role of epithelial NFκB in mediating innate inflammation in vivo has focused our studies on understanding its mechanism for gene activation. High throughput chromatin immunoprecipitation next generation sequencing (ChIP-Seq) have shown that RelA binds to ~5000 high affinity sequences in the genome, located primarily on cytokine, anti-apoptotic, and cell stress-response genes [49,50]. Intriguingly, time course experiments have shown that RelA- dependent gene expression occurs in temporally distinct waves, suggesting additional epigenetic regulation of target gene activity. The most rapidly activated wave of gene expression is comprised of cytokines and anti-apoptotic factors. The teleological explanation for this composition is that rapid expression of paracrine alarm signals is critical for host survival [51]. A major insight into the mechanisms how this group of immediate-early genes are activated was made with the discovery that phosphorylated NFκB interacts with the positive P-TEFb complex, a multifunctional gene regulatory complex that includes cyclin dependent kinase (CDK) 9 and bromodomain containing protein 4 (BRD4) as major effector enzymes [24,52,53]. Through its effect on RNA polymerase processivity and chromatin modifications, P-TEFb is a major NFκB coactivator in the IIR. We found that RSV-induced oxidative stress triggers NFκB/Rel Ser 276 phosphorylation, a posttranslational modification stoichiometrically coupled with Lys 310 acetylation recognized by bromodomain containing 4 (BRD4) in the P-TEFb complex [52].

CDK9 is found in a heterogeneous protein complex that exists in several states; one an inactive state bound with the abundant small nuclear RNA, 7SK snRNA, the hexamethylene bisacetamide-inducible proteins (HEXIM1/2) and the methyl phosphate capping enzyme (BCDIN3) [54,55], and the other, an active state associated with bromodomain containing 4 (schematically illustrated in Figure 2). In the process of P-TEFb activation, BRD4 is recruited from other genomic sites, where it serves as an adapter form the CDK9-acetylated NFκB complex. The CDK9•BRD4 complex is then redistributed to genomic targets through association with sequence specific DNA-binding activity of NFκB [24,56].

## 9. Mechanism of Transcriptional Elongation in the IIR

Although virtually all steps of transcriptional activation are regulated, immediate-early IIR genes are primarily controlled at the level of transcriptional elongation. Schematically shown in Figure 3, immediate early primary response genes are found in open chromatin configuration bound by hypophosphorylated RNA polymerase II (Pol II). Hypo-phosphorylated RNA Pol II is complexed with a negative elongation factor (NELF)•·5,6-dichloro-1-β-D-ribofuranosylbenzimidazole(DRB) Sensitivity Inducing Factor (DSIF) complex cycles nonproductively on the 5’ upstream of IIR genes, producing short noncoding (~30–50 nt) RNA transcripts [53,57]. When P-TEFb is recruited to these paused promoters, the CDK9•BRD4-dependent kinase activity phosphorylates Ser2 in the heptad repeats of the RNA Pol II carboxy terminal domain (CTD), as well as NELF·DSIF. NELF is evicted from the Pol II complex upon its phosphorylation and DSIF switches into a positive elongation factor [58,59,60]. The phospho-Ser 2 CTD Pol II DSIB complex acquires a processive function, enabling RNA Pol II to traverse the gene body and express full-length, fully spliced RNA transcripts. This model is supported by time series ChIP experiments that show CDK9 and Pol II complexes dynamically redistribute from the promoter to over the body of immediate early genes [52,53].

The essential functional role of CDK9•BRD4 in RSV-induced IIR has been demonstrated by disruption of P-TEFb recruitment by RelA Ser 276 site mutation [52], siRNA- mediated silencing of CDK9 and BRD4 [24,52,53,61], and small-molecule inhibition of CKD9 and BRD4 [34,48,53]. Consequently, CDK9•BRD4 are molecules receiving significant attention as a target for pharmacological manipulation of viral and allergen-inducible mucosal inflammation [48,62,63,64,65,66].

## 10. The IIR Induces Dynamic Changes in the P-TEFb Interactome and Its Gene Targets

Unbiased affinity enrichment studies have revealed remarkable functional diversity of the P-TEFb complex, where, in the unstimulated cell state, CDK9 is in a complex with over 560 cellular proteins and 7SK snRNA [55,67]. These proteins include structural ribosomal subunits and RNA splicing factors, including DDX-containing RNA helicase/RNA splicing proteins. Upon RIG-I activation, CDK9 binding to 7SK snRNA was replaced by 77 additional proteins; these proteins include chromatin modifying proteins, histones, peroxiredoxins, and signal transducers and activators of transcription (STATs) [67]. The presence of proteins important in RNA export, translation initiation, and chromatin remodeling proteins suggests much greater complexity of PTEFb complex that will require further investigation.

A causal network analysis of “modulators” (proteins that modify the activity of CDK9) and corresponding target genes underscores how the innate response shifts the activity of the P-TEFb complex to alter cellular functions [68]. In this analysis, we identified basal and inducible modulators of CDK9 activity by learning from over 2000 high throughput gene expression experiments in ExpO [67]. The inducible CDK9 modulator complex (CM2 in Figure 4) controls target gene networks important in hypoxic signaling, angiogenesis, stress and defense response, wound healing, and regulation of innate immune response [CDK9 target (CT)1, 2 in Figure 4]. This analysis indicates that the functions and gene targets of the CDK9 complex are dynamically shaped by the innate immune response.

## 11. BRD4 is a Pleiotrophic Coactivator of the P-TEFb Complex

Bromodomain containing protein 4 (BRD4) is a critical enzymatic component of the activated P-TEFb complex. Not only does BRD4 prevent the inhibitory HEXIM1/2•7SK snRNA from binding to CDK9 enabling the shift of P-TEFb from the inactive to the active complex, but BRD4 phosphorylates both CDK9 and RNA Pol II. BRD4 facilitates phosphorylation of RNA Pol II [70], regulating its enzymatic processivity and RNA splicing functions, resulting in the rapid expression of inflammatory genes [52,67].

Our unbiased systems level study of the BRD4 interactome indicates that BRD4 associates with ~1000 proteins, including RNA polymerases, enhancer binding proteins, spliceosomal proteins, and ATP-dependent chromatin remodeling proteins (Figure 5). Others have shown that BRD4 nucleates interactions with acetylated histones; BRD4’s affinity for binding diacetylated histone H4 (with preference for Lys 5,/8, 8/12 and 12/16), triacetylated H4 (Lys 12/16/20), and diacetylated H3 (on Lys 9/14) results in its primary distribution throughout the genome associated with H3K27Ac-enriched enhancers of actively expressed genes [71,72]. In addition, BRD4 bridges CDK9 binding to acetylated NFκB. Inducible complex formation with RelA restructures dense BRD4 foci (“super-enhancers”) away from differentiation-specific genes to inflammatory gene networks [73]. Finally, BRD4 functions as an atypical histone acetyltransferase mediating histone acetylation on Lys 122, promoting nucleosome destabilization, promoting transcriptional elongation [52,70,74,75,76,77,78].

We recently found that the RSV induced RelA•BRD4 association also induced the atypical histone acetyl transferase (HAT) activity of BRD4 and acetylating histone H3 on Lys (K) 122, a modification that destabilizes nucleosomes, enhancing transcription through GC-rich gene bodies [34,74]. To illustrate the inducible formation of RelA and BRD4 complexes in the airway mucosa in vivo, we conducted proximity ligation assays (PLAs). PLAs detect atomic-distance interactions that are detected by the enzymatic ligation of antibody (Ab)-selective oligonucleotides. In the absence of RSV infection, few BRD4•·RelA complex can be detected in the mouse airway. By contrast, after RSV infection, BRD4•RelA complex is strongly detected (Figure 6). Specificity of this interaction is shown by its loss in the RelA knockout in the *Scgb1a1* expressing epithelium. Finally, BRD4 promotes CDK9-dependent phosphorylation [70]. Our findings that global Ser2 CTD RNA Pol II activation is dependent on RelA suggest that formation of the RelA·BRD4 complex globally affects total nuclear RNA Pol II activity.

## 12. P-TEFb Plays a Role in Epithelial-Mesenchymal Transition (EMT)

Severe RSV LRTIs have been associated with persistent decreased pulmonary function and chronic obstructive pulmonary disease persisting into adulthood [80]. In chronic obstructive pulmonary disease (COPD), mucosal thickening and reduced small airway diameter are primarily responsible for reduced expiratory airflow. It is important to re-emphasize that RSV replicates in small airways. RSV replication has been documented in bronchioles and alveolar epithelial cells of children with RSV LRTI [2]. Discussed earlier, RSV challenge studies in normal immune volunteers demonstrated persistent viral replication and inflammation in the lower airways [9]. Of mechanistic relevance, our secretome analysis found that TGFβ and extracellular matrix proteins are abundantly secreted by RSV-infected bronchiolar cells [37]. Mechanistically, TGFβ receptor (TGFβRII) signaling activates epithelial mesenchymal transition (EMT) through Smad-dependent and Smad-independent pathways [78]. TGFβ-induced EMT leads to disruption of mucosal barrier function by inducing the loss of apical polarity, reduced epithelial cadherin (ECad), and disruption of epithelial adherens junctions. In addition, type II EMT enables transformed epithelial cells to express α-SMA stress fibers and intermediate filament vimentin, to produce extracellular matrix (ECM) through secretion of collagen and fibronectin, and to increase expression of matrix metalloproteinases (MMPs) to promote airway remodeling [81,82,83]. Although the mechanisms controlling RSV-induced remodeling are still under investigation, it has been clearly established that repeated viral pattern exposures induce airway mucosal changes associated with epithelial de-differentiation and mesenchymal transition in a manner similar to TGFβ [34,63].

Studies in *Scgb1a1+*-derived epithelial cells has shown that TGFβ- activates NFκB activity, and the transition is mediated by NFκB-dependent gene networks [83,84]. Here, NFκB/RelA is required for expression of master regulators of EMT, including Wingless-related integration site (WNT), Jun Proto-Oncogene(JUN), Snail family transcriptional repressor (SNAI1) and Zinc Finger E-Box Binding Homeobox (ZEB1) [82]. This has led to the exciting finding that repetitive stimulation of NFκB/RelA via the innate pathway reprograms the expression of fibrogenic mesenchymal genes [85]. As with the rapid activation of immediate-early IIR genes, NFκB binding recruits BRD4 to the *SNAI/ZEB1/TWIST* promoters and mediates ECM production and airway fibrosis [61]. Several studies have shown that small molecule BRD4 inhibitors block mesenchymal transition [16,85].

Collectively these studies show that the NFκB·BRD4 complex is a central mediator of innate inflammation-mediated mesenchymal transition. Acute activation of NFκB·BRD4 produces chemokine secretion and neutrophilic inflammation, whereas persistent activation triggers reprogramming of fibrogenic genes and the core transcriptional regulators of epithelial cell state transition.

## 13. Mesenchymal Transitioned Epithelial Cells Release Paracrine Factors that Expand Myofibroblast Population

Under normal conditions, epithelial cells provide trophic signals to an attenuated sheath of sub-epithelial mesenchymal cells, forming the epithelial–mesenchymal unit (EMU). Integrity of the EMU depends on growth factors secreted from epithelial cells [86]. Growth factor secretion is rapidly increased in response to injury or infection-induced EMT to promote regeneration of this critical mucosal surface. These growth factors include TGFβ, epidermal growth factor (EGF) connective tissue growth factor (CTGF), and fibrogenic cytokines (IL-6, TSLP, IL-33, IL-25, and others). In airway infection and injury, sub-epithelial fibroblasts, circulating fibrocytes and perivascular pericytes are transitioned to αSMA and COL1-expressing myofibroblasts. αSMA^+^/COL1^+^- co-expressing myofibroblasts are a secretory phenotype of lung stromal mesenchymal cells that are major producers of ECM proteins and MMPs that contribute to lamina reticularis expansion [87]. We have observed that RSV infection dynamically expands the myofibroblast population, demonstrated by the increased population of αSMA^+^/COL1^+^ cells underneath the epithelial layer (Figure 7). Importantly, this myofibroblast transition is dependent on BRD4, since the population is not observed in RSV infected mice treated with a nonselective BRD4 inhibitor.

A collective view of the relationship between innate signaling, mesenchymal transition and myofibroblast expansion is shown in Figure 8. In addition to the anti-viral response, Th2 polarizing activities, and mucogenic responses, RSV-infected epithelial cells also promote ECM deposition, MMP expression, and expansion of the myofibroblast population. More studies will need to be completed to determine if this paracrine myofibroblast pathway mediates long term airway remodeling in severe RSV-induced LRTI.

## 14. Discussion and Future Directions

RSV is an important human pathogen that produces recurrent viral infections throughout life. In this review, I have summarized some of the recent work that has focused on the unique properties of the bronchiolar cells derived from *Scgb1a1*-expressing progenitors. These cells are an important sentinel cell for viral replication in the airway. In contrast to epithelial cells from the conducting airways, this population of cells synthesize and secrete Th2-polarizing, mucogenic, and profibrotic cytokines in response to NFκB activation. These different chemokine expression patterns are immunologically significant and play an important role in the pathogenesis of LRTIs, including airway remodeling. NFκB functions as a master regulator of the innate response controlling IFN pathway. This paper reviews our understanding how NFκB as a master regulator of TGFβ- and viral PAMP-induced mesenchymal transition. A key finding has been that NFκB associates with P-TEFb and repositions it to innate and fibrogenic gene networks.

An important area for future investigation include understanding the diverse functions of the P-TEFb complex. Both CDK9 and BRD4 interact with proteins involved in RNA splicing, export, and translational initiation. These intriguing findings strongly indicate that P-TEFb is a multifunctional coactivator complex that may control transcription, splicing and protein synthesis of immediate-early genes. The roles of the late-induced gene network in RSV disease and its sequelae are currently unknown. Much more exciting work remains to be done to understand how this pathway may be involved in Th2 polarization, myofibroblast transition, and/or remodeling. Moreover, because small molecule regulators of the P-TEFb complex have been developed opens new translational approaches to inhibit NFκB-BRD4 signaling that may reverse mesenchymal transition, restore disrupted barrier function, and reduce loss of lung function after RSV LRTI.

## Figures and Tables

**Figure 1 viruses-12-00472-f001:**
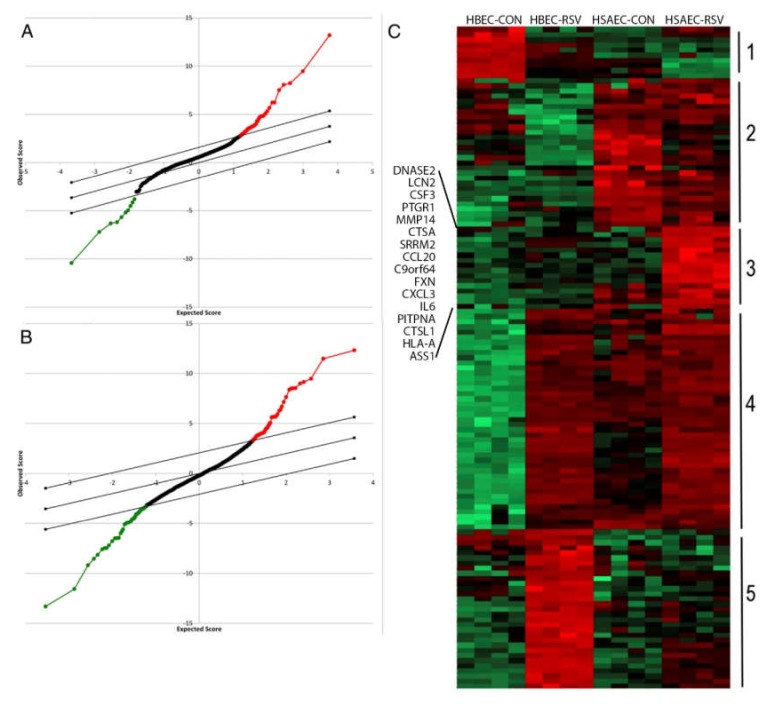
Identification of differentially secreted proteins in response to respiratory syncytial virus (RSV) infection. (**A**). Statistical analysis of microarray (SAM) for secretome proteins whose expression differs by cell type in the basal state. X axis, expectation score; Y axis, observation score. The diagonal line shows where false discovery rate = 0.01. Points above (in red) outside the threshold are those with FDR < 0.01, and points below the threshold (in green) are those proteins with FDR > 0.01. Proteins with increased expression in hSAECs are indicated by red points; those decreased are indicated in green. (**B**). SAM for secretome proteins whose expression differs due to RSV infection. (**C**). Expression values of proteins were z-score-normalized data of log2-transformed expression values for biological replicates. Two-dimensional hierarchical clustering was performed, with columns representing cell samples and rows representing individual proteins (green, low expression, red, high expression). Cluster 3 is unique to the bronchiolar-derived human small airway epithelial cells (HSAECs). HSAECs differentially express CCL20, TSLP, IL6, and MIP1α. Reproduced with permission from [37].

**Figure 2 viruses-12-00472-f002:**
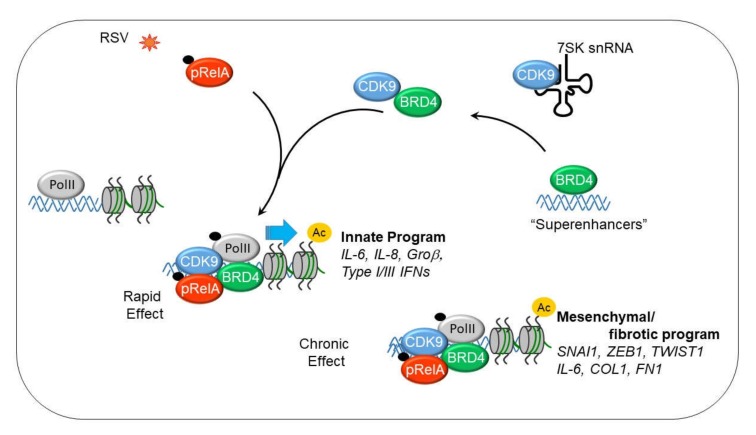
Dynamic changes in P-TEFb complex in response to innate immune response (IIR). Shown is a schematic view of the nuclear states of activated P-TEFb. Approximately half of nuclear P-TEFb is inactive state associated with HEXIM1 and 7SK RNA. The component that is actively involved in RNA Pol II dependent transcription is associated with bromodomain containing 4 (BRD4). BRD4 is recruited from genomic sites, including super enhancers that control cell type specific gene expression. BRD4 serves as an adaptor to bind phosphorylated/acetylated NFκB/RelA to recruit to innate response genes engaged with hypophosphorylated RNA Pol II. cyclin dependent kinase 9 (CDK9) phosphorylates Ser2 in the heptad repeat of the carboxyterminal domain (CTD) of RNA polymerase II (Pol II), acutely activating the IIR and chronically activating mesenchymal transition.

**Figure 3 viruses-12-00472-f003:**
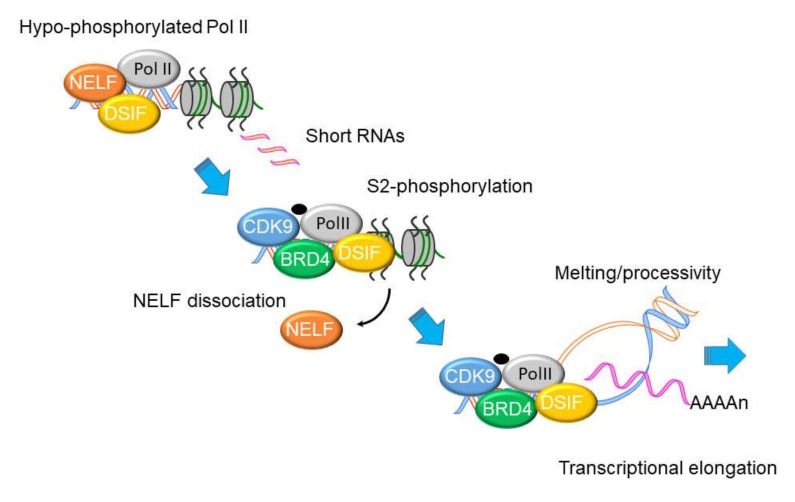
Molecular steps in inducible transcriptional elongation. Genes in the innate immune response are in open chromatin configuration engaged with hypophosphorylated RNA Polymerase II (Pol II) and complexed with negative elongation factor (NELF) and DRB sensitivity inducing factor (DSIF). Only short, unprocessed RNAs are produced. With CDK9•BRD4 recruitment, the carboxy terminal domain (CTD) of RNA Poll II is phosphorylated, resulting in disruption of NELF binding. Activated RNA Pol II becomes processive, producing full length, polyadenylated RNA transcripts.

**Figure 4 viruses-12-00472-f004:**
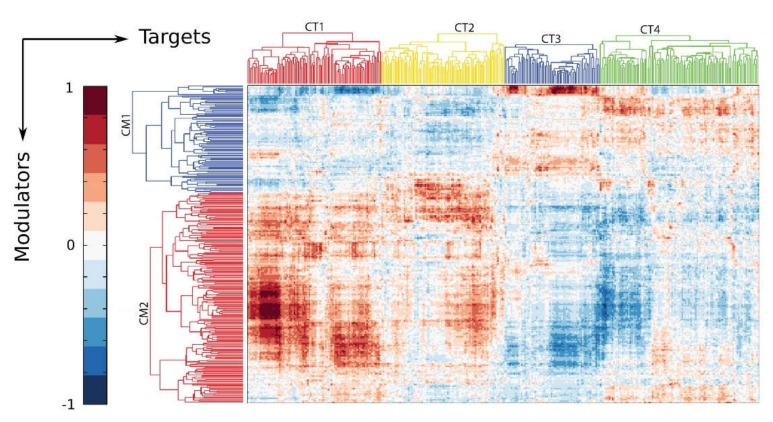
Innate activation affects the global CDK9 modulatory network. The CDK9 modulatory network was constructed from all known CDK9 binding proteins as candidate modulators and all target gene candidates. Shown is a biclustering heatmap. Each row represents a modulator probe set, each column represents a target gene probe set and each element represents the γ parameter of the modulator-target gene (TG) model [69]. Proteins induced to bind CDK9 are enriched in the modulator cluster 2 (CM2), and associate with target genes in cluster CT-1 and -2.

**Figure 5 viruses-12-00472-f005:**
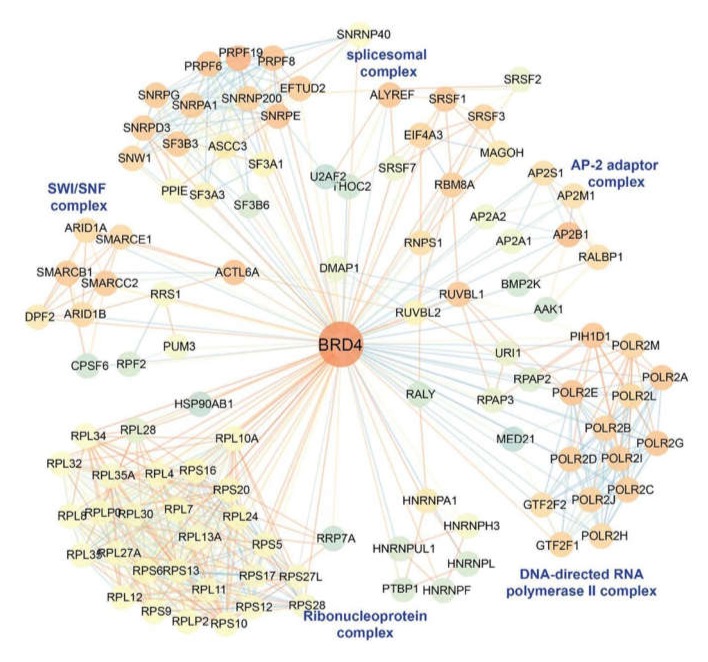
BRD4 interactome. Unbiased immunoprecipitation-mass spectrometry (IP-MS) analysis was used to determine the BRD4 interactome. The color of each node represents the neighborhood connectivity of each protein. The color of edges corresponds to the MiST score from interactions between RelA and its interactors and also to interactions between RelA interactors that were obtained from publicly available protein-protein interaction (“STRING”) database. Reproduced with permission from [79].

**Figure 6 viruses-12-00472-f006:**
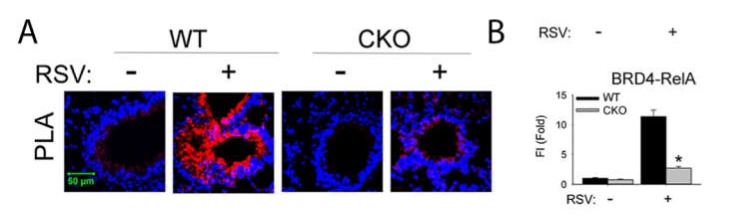
Inducible NFκB•BRD4 complex formation *i**n vivo*. **(****A),** PLA assays of RelA–BRD4 molecular interactions were performed with lung sections from oil- or tamoxifen (TMX)-treated *Scgb1a1*Cre ^ERTM^/^+^ × RelA^fl/fl^ mice in the absence or presence of RSV. Foci of interactions are amplified as red foci; sections are counterstained with 4′,6-diamidino-2-phenylindole (DAPI, blue). CKO, cre-mediated knockout of RelA; WT, wild type. Magnification, ×63. **(****B),** quantifications of data from the proximity ligation assays (PLA). Data from [47].

**Figure 7 viruses-12-00472-f007:**
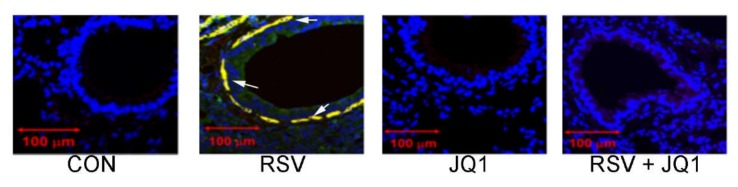
RSV induced sub-eptihelial myofibroblasts is BRD4 dependent. Shown is immunohistochemistry from mouse airway +/- RSV infection, in the absence or presence of the nonselective thienotriazolodiazepine BRD4 inhibitor, JQ1 5 d after RSV infection. Co-staining of the aSMA+/COL1 expressing myofibroblasts are shown in yellow and indicated by arrows. CON, uninfected. Data from [45].

**Figure 8 viruses-12-00472-f008:**
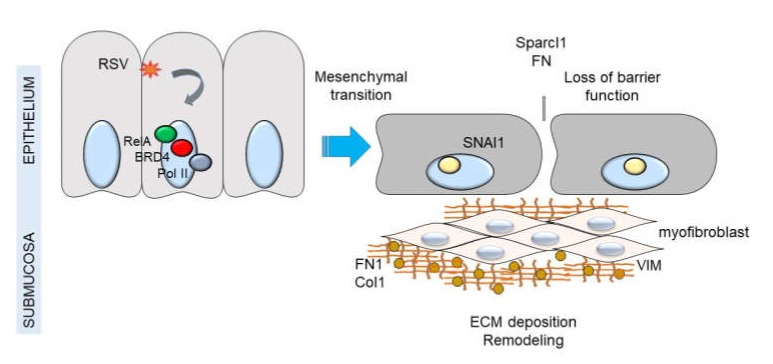
NFκB-BRD4 complex links RSV-induced IIR with airway remodeling. Shown is a schematic model of airway epithelium responding to viral pathogen associated molecular patterns (PAMPs). Either persistent activation of the NFκB pathway by TLR3 or RIG-I results in remodeling the PTEFb complex and association with NFκB/RelA. Persistent activation of this pathway results in up-regulation of the core EMT regulator SNAI1, as well as enhanced production of extracellular matrix proteins FN1 and Col1A. Expansion of myofibroblast population is indicated by interstitial VIM up-regulation and accumulation of hydroxyproline. Down-regulation of CDH1 results in reduced adherens junctions and enhanced epithelial permeability.

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
