# Peer review of "RSV Reprograms the CDK9•BRD4 Chromatin Remodeling Complex to Couple Innate Inflammation to Airway Remodeling"

_viruses, 2020, doi:10.3390/v12040472_

Round 1

Reviewer 1 Report

In this review, Dr. Brasier summarized how the CDK9/BRD4 chromatin remodeling complex couple innate inflammation to remodel the airway in the context of RSV infection. The description flows smoothly and well summaries years of work from his laboratory. No major concerns, but including plans in the discussion would be highly appreciated. For example, is the team going to take a look at Th2 or CD8 cell responses in their airway remodeling model? Are they altered by CDk9/BRD4? etc.

Minor errors:

Line 37.. space between (RSV) and is.

Line 56, CD*T cells should be CD8T cells.

Lines 92,95, 98,189,201,226,242, 358, 361, and 399, kappa is missing

Line 131 missing alpha after MIP

Line 143 these to “These”.

Lines 144. 199, 211, Need to change to bold

The resolution of Fig 1 needs to be improved. It is hard to read.

For the section of “9. Mechanism of transcriptional elongation in the IIR”, it would be better to include another figure to interpret the interactions among the molecules.

Figure 6. C and D should be “A and B”.

Author Response

Thank you for the comments, suggestions and corrections.  I added a Discussion and Future Directions section that addresses next steps. 

Minor errors:

We apologize for the loss of font style in conversion of the word document and have corrected the typographical errors identified in the revised manuscript. 

In Fig. 1, I have increased the font size of the proteins identified in cluster 3, the major focus of this paper.  The identity of the other proteins are found in Fig. 10 in the original paper from which this figure was derived (ref [35]). 

I have added a new figure that describes the molecular steps in transcription elongation, replacing Fig. 3 in the revised manuscript. 

Labeling on panels of Fig 6 have been corrected.

Reviewer 2 Report

Minor spell check

L37: (RSV)is

L56: memory CD*+ T cells

L66: “URIs” is not defined (upper respiratory infections I guess)

L95: proteolysis of I  B  inhibitors

L98: innate NF B-dependent

L106: title is not in bold

L143: these

L144: title is not in bold

L161: many of these

L169: NF B (In my pdf version, I think the “kappa” disappeared most of the time)

L189: NF B

L194: hSAECs is not defined (human small airway epithelial cells I guess)

L197: paracein

L199: title is not in bold

L201: NF B

L211: title is not in bold

L226: NF B

Figure 3: This figure is not clear and seems unnecessary. There are 16 different “class” for the Genome Ontology for 17 bars. The differential network is unreadable.

L294: the     parameter

L329: Finally

Figure6: CKO should be defined in the figure legend

L333: NF B

L337: reproduced with permission from (45)

L358: NF B

L361: NF B

L366-367: NF B

L377: (EGF),

L380: to      SMA (alpha-SMA I guess)

Figure 7: could you precise the time after the RSV infection to observe subepithelial myofibroblast please?

L390: co-staining

L391: reproduced with permission from (45)

L399-421: NF B

Questions

1- Regarding airway remodeling, you mention EMT, ECM deposition and myofibroblasts, what about bronchial smooth muscle remodeling in asthma? Is RSV infection linked to bronchial smooth muscle proliferation?

2- The author mentions the infection of airway epithelial cells, what about airway macrophages? They also represent a first line of defense and are infected by RSV. Do we have informations of the role of RSV-infected airway macrophages in airway remodeling?

3- Do we have an idea if this mechanisms or chromatin remodeling leading to airway remodeling might be different between adults and children? This is something that may be discussed regarding future therapeutic strategies.

Author Response

thank you for the feedback on the manuscript. 

I have removed Fig. 3 as suggested. 

Figure 6: CKO is defined in the figure legend (CRE-mediated knockout)

Figure 7: Images are obtained after 5 d of RSV infection- added to figure legend.

Responses to specific questions: 

Is RSV infection linked to bronchial smooth muscle proliferation?  I don’t think this is known. Smooth muscle hypertrophy has not been observed in human histology from fatal cases. 

Do we have informations of the role of RSV-infected airway macrophages in airway remodeling?  there is less information on the role of alveolar macrophages.  Some studies using macrophage depletion by the Garofalo group indicate that macrophages modulate the inflammatory and disease response.  This information is stated and publication cited on line 131. 

Do we have an idea if this mechanisms or chromatin remodeling leading to airway remodeling might be different between adults and children?  This is an interesting question, and could be focus of future investigations.